# RESTORER GUIDED DIFFUSION MODELS FOR VARIATIONAL INVERSE PROBLEMS

## ABSTRACT

Diffusion models have made remarkable progress in solving various inverse problems, attributing to the generative modeling capability of the data manifold. Posterior sampling from the conditional score function enable the precious data consistency certified by the measurement-based likelihood term. However, most prevailing approaches confined to the deterministic deterioration process of the measurement model, regardless of variational unpredictable disturbance in real-world sceneries. To address this obstacle, we show that the measurement-based likelihood can be replaced with restoration-based likelihood in the opposite probabilistic graphic direction, licencing the patronage of various off-the-shelf restoration models and extending the strict deterministic deterioration process to the tolerant cluster process with supposed prototype, in what we call restorer guidance. Particularly, assembled with versatile prototypes optionally, we can resolve inverse problems with bunch of choices for assorted sample quality and realize the proficient deterioration control with assured realistic. We show that our work can be formally analogous to the transition from classifier guidance to classifier-free guidance in the field of inverse problem solver. Experiments on multifarious inverse problems demonstrate the effectiveness of our method, including image dehazing, rain streak removal, and motion deblurring. Code will be available soon.

## 1 INTRODUCTION

> *"Mille viae ducunt homines per saecula Romam."*
>
> *Liber Parabolarum Ālani*

Diffusion models Sohl-Dickstein et al. (2015); Ho et al. (2020); Song et al. (2020) have recently emerged as impressive generative models with promising performance on various applications such as image generation Rombach et al. (2022); Zhang & Agrawala (2023); Saharia et al. (2022), image editing Meng et al. (2021); Brooks et al. (2023); Ruiz et al. (2023), video generation Ho et al. (2022), speech synthesis Huang et al. (2022), and 3D generative modeling Poole et al. (2022); Tewari et al. (2023). Apart from that, diffusion models are also served as competitive candidates for inverse problem solver, which aim at reversing the deterioration process from the contaminated measurement $y$ to original complete signal $x$ Chung et al. (2022; 2023b); Song et al. (2023).

Solving inverse problems with diffusion models can be crafted in multiform frameworks. Bayesian approach incorporates the gradients from the measurement-based likelihood, i.e., $\nabla_{x} \log p(y|x)$, forming the conditional score function for posterior sampling, and the data consistency can be ensured with the dependency derived from the measurement model $\mathcal{H}$. Representative methods Chung et al. (2022; 2023b); Song et al. (2023) progressively extend the diffusion solvers with linear, nonlinear, or even non-differentiable measurement models for increasingly complicated inverse problems. Beyond the Bayes' formula, there are broad range of alternatives delivering the balance between data fidelity and realistic for solving inverse problems, such as range-null space decomposition Wang et al. (2023) and heuristic energy function with configured properties Fei et al. (2023); Zhao et al. (2022). These methods can be comfortably adapted to multifarious inverse problems without retraining the diffusion model. However, it is worth noting that most prevailing approaches confined to the deterministic deterioration process of the measurement model, mostly involving the digitized deterioration such as image inpainting, image colorization, and phase retrieval, regardless of variational unpredictable disturbance in real-world sceneries, including but not limited to variational weather conditions Zhu et al. (2023) or manual destruction Köhler et al. (2012).

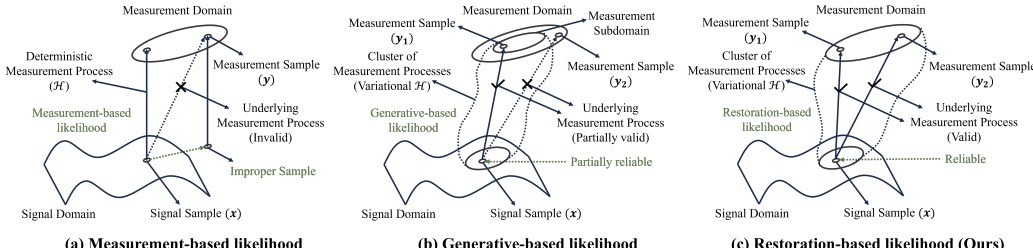

Figure 1: Visual illustration of various likelihood terms in prevailing diffusion-based inverse problem solvers. Compared to the deterministic deterioration process of the measurement-based likelihood, the generative-based and restoration-based likelihood are capable of handling variational deterioration process with reliable likelihood derived from the congruous deterioration process and the measurement, while the generative-based likelihood further restricted to the rigid formulation.

Another line of works Stevens et al. (2023); Chung et al. (2023a) introduce the generative-based likelihood with parallel diffusion models for signal $x$ and deterioration parameters in measurement model $\mathcal{H}$, and jointly estimate their score functions for posterior sampling, which release the deficiency of the deterministic deterioration process with bestowed variational capability. Additionally, Laroche et al. (2023) alternately estimates the measurement parameters and data distribution under the traditional iterative optimization framework in the same spirit. However, these methods remain in the paradigm of the measurement-based likelihood, and confined to the rigid formulation of the measurement model for signal formation, a.k.a., convolution, addition, and multiplication, with merely estimated deterioration parameters, which inevitably restricts their variational capability for more complicated sceneries. Moreover, it is noteworthy that aside from the aforementioned *pros.* and *cons.* of various likelihood terms, the coupled learning of the measurement model is necessary to be realized on-the-fly, which is substantially time-consuming and inconvenient to deploy.

In this work, we extend prevailing diffusion solvers for variational inverse problems beyond the restriction of deterministic deterioration process without any extra training. In the context of Bayes' framework, we show that the measurement-based likelihood can be replaced with restoration-based likelihood in opposite probabilistic graphic direction, forming the reliable conditional score function for posterior sampling, in what we call *restorer guidance*. Compared with measurement-based likelihood, *restorer guidance* licences the patronage of various off-the-shelf restoration models, and implicitly extends the strict deterministic process in measurement-based likelihood to a cluster of deterioration processes with supposed restorer prototype for variational inverse problem solver. In Fig. 1, we further illustrate that the devil in measurement-based likelihood resides in the incongruous dependency between the forward deterioration process and the contaminated measurement, which can be properly resolved with tolerant cluster process derived from the restorer prototype for reliable likelihood. Assembled with versatile restorer prototypes optionally, we can resolve inverse problems with bunch of choices for assorted sample quality and realize the proficient deterioration control with assured realistic. We show that our work can be formally analogous to the transition from classifier guidance Dhariwal & Nichol (2021) to classifier-free guidance Ho & Salimans (2022) in the field of inverse problem solver. Note that our method is also compatible with other frameworks beyond Bayesian, such as range-null space decomposition (see Appendix B).

Empirically, we demonstrate the effectiveness of our method on various variational inverse problems, including image dehazing, rain streak removal, and motion deblurring, and show that our *restorer guidance* is a competitive inverse problem solver. The *restorer guidance* is not only capable of exploiting the restoration capability conserved in restorers losslessly, but rather breaking the upper bound of the restorer for superior sample quality (Fig. 3). Moreover, *restorer guidance* is also favourable to the out-of-distribution deterioration with augmented cluster process.

## 2 BACKGROUND

### 2.1 SCORE-BASED DIFFUSION MODELS

Score-based diffusion models smoothly transform data distribution to spherical Gaussian distribution with a diffusion process, and reverse the process with score matching to synthesize samples. The *forward process* $\{x(t)\}_{t \in [0,T]}$, $x(t) \in \mathbb{R}^D$, can be represented with the following Itô stochastic

differential equation (SDE) Song et al. (2020):
$$d\boldsymbol{x} = \boldsymbol{f}(\boldsymbol{x}, t)dt + g(t)d\boldsymbol{w}, \tag{1}$$
where $\boldsymbol{f}(\cdot, t) : \mathbb{R}^D \to \mathbb{R}^D$ is the drift coefficient, $g(t) \in \mathbb{R}$ is the diffusion coefficient, and $\boldsymbol{w} \in \mathbb{R}^D$ is the standard Wiener process (a.k.a., Brownian motion). Let $p_t(\boldsymbol{x})$ denotes the marginal distribution of $\boldsymbol{x}(t)$. The data distribution is defined when $t = 0$, i.e. $\boldsymbol{x}(0) \sim p_{\text{data}}$, and the tractable prior distribution is approximated when $t = T$, e.g. $\boldsymbol{x}(T) \sim \mathcal{N}(\boldsymbol{0}, \boldsymbol{I})$. $p_{0t}(\boldsymbol{x}_t|\boldsymbol{x}_0)$ denotes the transition kernel from $\boldsymbol{x}(0)$ to $\boldsymbol{x}(t)$. Note that we always have $p_0 = p_{\text{data}}$ by forward definition 1.

Samples from $p_t(\boldsymbol{x})$ can be simulated via the associated *reverse-time diffusion process* of 1, solving from $t = T$ to $t = 0$, given by the following SDE Anderson (1982); Song et al. (2020)
$$\mathrm{d}\boldsymbol{x} = [\boldsymbol{f}(\boldsymbol{x}, t) - g(t)^2 \nabla_{\boldsymbol{x}} \log p_t(\boldsymbol{x})]\mathrm{d}t + g(t)\mathrm{d}\overline{\boldsymbol{w}}, \tag{2}$$
where $\overline{\boldsymbol{w}}$ is the reverse-time standard Wiener process, and $\mathrm{d}t$ is an infinitesimal negative timestep. The reverse process of 2 can be derived with the *score function* $\nabla_{\boldsymbol{x}} \log p_t(\boldsymbol{x})$ at each time $t$, which is typically replaced with $\nabla_{\boldsymbol{x}(t)} \log p_{0t}(\boldsymbol{x}(t)|\boldsymbol{x}(0))$ in practice, and is approximated via score-based model $\boldsymbol{s}_\theta(\boldsymbol{x}(t), t)$ trained with *denoising score matching objective* Vincent (2011):
$$\theta^* = \arg\min_{\theta} {}_{t \sim U(\varepsilon, 1), \boldsymbol{x}(t) \sim p_{0t}(\boldsymbol{x}(t)|\boldsymbol{x}(0)), \boldsymbol{x}(0) \sim p_{\text{data}}} \left[ \|\boldsymbol{s}_\theta(\boldsymbol{x}(t), t) - \nabla_{\boldsymbol{x}(t)} \log p_{0t}(\boldsymbol{x}(t)|\boldsymbol{x}(0))\|_2^2 \right], \tag{3}$$
where $\varepsilon \simeq 0$ is a small positive constant. Score matching 3 ensure the optimal solution $\theta^*$ converges to $\nabla_{\boldsymbol{x}} \log p_t(\boldsymbol{x}) \simeq \boldsymbol{s}_{\theta^*}(\boldsymbol{x}(t), t)$ with sufficient data and model capability. One can replace the score function in 2 with $\boldsymbol{s}_{\theta^*}(\boldsymbol{x}_t, t)$ to calculate the *reverse-time diffusion process* Song et al. (2020) and solve the trajectory with numerical samplers, such as Euler-Maruyama, Ancestral sampler Ho et al. (2020), probability flow ODE Song et al. (2020), DPM-Solver Lu et al. (2022), amounts to sampling from the data distribution $p_{\text{data}}(\boldsymbol{x})$ with the goal of generative modeling.

## 2.2 Solving Inverse Problem with Diffusion Models

Solving inverse problem with diffusion model leverage the implicit prior of the underlying data distribution that the diffusion model have been learned Chung et al. (2022; 2023b); Song et al. (2023); Stevens et al. (2023). Formed in the Bayes' framework, we have $p(\boldsymbol{x}|\boldsymbol{y}) = p(\boldsymbol{y}|\boldsymbol{x})p(\boldsymbol{x})/p(\boldsymbol{y})$. Let $\boldsymbol{y}$ denotes the contaminated observation derived from the complete measurement $\boldsymbol{x}$, we can straightforward modify the unconditional score function in 2 with the following posterior formula, which similar to the classifier guidance Dhariwal & Nichol (2021):
$$\nabla_{\boldsymbol{x}_t} \log p_t(\boldsymbol{x}_t|\boldsymbol{y}) = \nabla_{\boldsymbol{x}_t} \log p_t(\boldsymbol{x}_t) + \nabla_{\boldsymbol{x}_t} \log p_t(\boldsymbol{y}|\boldsymbol{x}_t), \tag{4}$$
where the prior term can be approximated via the pre-trained score model $\boldsymbol{s}_{\theta^*}(\boldsymbol{x}_t, t)$, and the likelihood term can be acquired via the compound of the Tweedie's formula Efron (2011) and the measurement model from $\boldsymbol{x}$ to $\boldsymbol{y}$ to ensure the data consistency. Simply replacing the score function in 2 with 4 enable the conditional *reverse-time diffusion process* for posterior sampling:
$$\mathrm{d}\boldsymbol{x} = \left[ \boldsymbol{f}(\boldsymbol{x}, t) - g(t)^2 (\nabla_{\boldsymbol{x}_t} \log p_t(\boldsymbol{x}_t) + \nabla_{\boldsymbol{x}_t} \log p_t(\boldsymbol{y}|\boldsymbol{x}_t)) \right] \mathrm{d}t + g(t)\mathrm{d}\overline{\boldsymbol{w}}, \tag{5}$$
where the first term promise the realistic powered by diffusion manifold constraint, and the second term ensure the data fidelity. It is worth noting that the likelihood can be further approximated with heuristic energy function with configured properties Zhao et al. (2022); Fei et al. (2023).

## 3 Methods

### 3.1 Approximating the measurement-based likelihood

Recall that the posterior sampling from the conditional score function 5 require the likelihood term $\nabla_{\boldsymbol{x}_t} \log p_t(\boldsymbol{y}|\boldsymbol{x}_t)$ to provide the guidance which is intractable to compute. Pioneer works typically factorize $p_t(\boldsymbol{y}|\boldsymbol{x}_t)$ with the marginalization over $\boldsymbol{x}_0$, considering the underlying graphic model:
$$p(\boldsymbol{y}|\boldsymbol{x}_t) = \int_{\boldsymbol{x}_0} p(\boldsymbol{y}|\boldsymbol{x}_0, \boldsymbol{x}_t)p(\boldsymbol{x}_0|\boldsymbol{x}_t)d\boldsymbol{x}_0 = \int_{\boldsymbol{x}_0} p(\boldsymbol{y}|\boldsymbol{x}_0)p(\boldsymbol{x}_0|\boldsymbol{x}_t)d\boldsymbol{x}_0, \tag{6}$$
Note that $\boldsymbol{x}_t$ is independent of the measurement $\boldsymbol{y}$ when conditioned on $\boldsymbol{x}_0$. In this way, we can accordingly approximating the $p(\boldsymbol{x}_0|\boldsymbol{x}_t)$ via one-step denoising process with Tweedie's formula Efron (2011), and solving the $p(\boldsymbol{y}|\boldsymbol{x}_0)$ from the measurement model. Unfortunately, the prevalent measurement-based likelihood is restricted to the deterministic deficiency of the measurement model, impeding the diffusion solvers for variational inverse problems; detailed in Appendix D.

## 3.2 RESTORER GUIDANCE

To address the abovementioned limitations, we show that the measurement-based likelihood can be replaced with restoration-based likelihood for data consistency, in what we call *restorer guidance*. Compared with measurement-based likelihood, the *restorer guidance* licencing the patronage of various off-the-shelf restoration models for powerful diffusion solvers, considering their comprehensive sensitivity to multifarious deterioration process. We first write the factorized restoration-based likelihood $\hat{p}(\boldsymbol{x}_t|\boldsymbol{y})$ as the following for comparison, and the modified conditional score function together with the restorer guided posterior sampling will be introduced later.

$$\hat{p}(\boldsymbol{x}_t|\boldsymbol{y}) = \int_{\boldsymbol{x}_0} p(\boldsymbol{x}_t|\boldsymbol{x}_0, \boldsymbol{y})p(\boldsymbol{x}_0|\boldsymbol{y})d\boldsymbol{x}_0 = \int_{\boldsymbol{x}_0} p(\boldsymbol{x}_t|\boldsymbol{x}_0)p(\boldsymbol{x}_0|\boldsymbol{y})d\boldsymbol{x}_0, \tag{7}$$

where the measurement $\boldsymbol{y}$ is independent of $\boldsymbol{x}_t$ when conditioned on $\boldsymbol{x}_0$. Note that the probabilistic graphic direction of Eq. 7 is opposite to the measurement-based likelihood (Eq. 6) for confident data consistency, as shown in Fig. 2. Solving $p(\boldsymbol{x}_0|\boldsymbol{y})$ with assorted restoration models $\mathcal{R}$ enable the establishment of variational cluster process. While the $p(\boldsymbol{x}_t|\boldsymbol{x}_0)$ can be directly derived from the forward process, e.g., $p(\boldsymbol{x}_t|\boldsymbol{x}_0) \sim \mathcal{N}(\sqrt{\bar{\alpha}(t)}\boldsymbol{x}_0, (1 - \bar{\alpha}(t))\boldsymbol{I})$, in the case of VP-SDE or DDPM Ho et al. (2020). Therefore, we have $\hat{p}(\boldsymbol{x}_t|\boldsymbol{y}) \sim \mathcal{N}(\sqrt{\bar{\alpha}(t)}\mathcal{R}(\boldsymbol{y}), (1 - \bar{\alpha}(t))\boldsymbol{I})$, considering the deterministic process of $p(\boldsymbol{x}_0|\boldsymbol{y})$. The score of the restoration-based likelihood can be written as:

**Measurement-based likelihood**

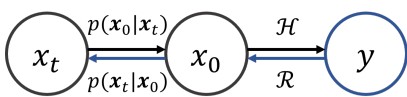

**Restoration-based likelihood**

Figure 2: Probabilistic graphic model. The direction of restoration-based likelihood is opposite to the prevailing measurement-based likelihood.

$$\nabla_{\boldsymbol{x}_t} \log \hat{p}(\boldsymbol{x}_t|\boldsymbol{y}) \simeq -\frac{1}{\sigma_t^2}\nabla_{\boldsymbol{x}_t}\|\boldsymbol{x}_t - \sqrt{\bar{\alpha}(t)}\mathcal{R}(\boldsymbol{y})\|_2^2 \tag{8}$$

where $\sigma_t$ is exactly the standard deviation of $p(\boldsymbol{x}_t|\boldsymbol{y})$, and we discard it to transform the underlying distribution of the mean-reverting error (Eq. 8) from time-constant $\epsilon_t \sim \mathcal{N}(\boldsymbol{0}, \boldsymbol{I})$ to time-dependent $\epsilon_t \sim \mathcal{N}(\boldsymbol{0}, \sigma_t^2)$ for adaptive restorer guidance with relaxation related to the noise schedule. Another perspective is provided in Appendix A. Once we obtain the $\nabla_{\boldsymbol{x}_t} \log p(\boldsymbol{x}_t|\boldsymbol{y})$, we can freely plug it into the modified conditional score function for restorer guided posterior sampling.

## 3.3 POSTERIOR SAMPLING FROM RESTORER GUIDANCE

To enable the posterior sampling from the restorer guidance and forming the branded conditional score function, we rewrite the likelihood term in Eq. 4 as following via Bayes' rule:

$$\nabla_{\boldsymbol{x}_t} \log p_t(\boldsymbol{y}|\boldsymbol{x}_t) = \nabla_{\boldsymbol{x}_t} \log \hat{p}_t(\boldsymbol{x}_t|\boldsymbol{y}) - \nabla_{\boldsymbol{x}_t} \log p_t(\boldsymbol{x}_t), \tag{9}$$

which translates the measurement-based likelihood $\nabla_{\boldsymbol{x}_t} \log p_t(\boldsymbol{y}|\boldsymbol{x}_t)$ to restoration-based likelihood $\nabla_{\boldsymbol{x}_t} \log \hat{p}_t(\boldsymbol{x}_t|\boldsymbol{y})$, where the resulting $\nabla_{\boldsymbol{x}_t} \log \hat{p}_t(\boldsymbol{y}|\boldsymbol{x}_t)$ is then used in $\nabla_{\boldsymbol{x}_t} \log p_t(\boldsymbol{y}|\boldsymbol{x}_t)$ when posterior sampling from diffusion solvers. Therefore, the conditional score function can be simply accessed by plugging in the derivation from Eq. 9 to Eq. 4. Considering the typical parameters $w$ that controls the strength of the measurement-based guidance, i.e., $w\nabla_{\boldsymbol{x}_t} \log p_t(\boldsymbol{y}|\boldsymbol{x}_t)$, we have:

$$\nabla_{\boldsymbol{x}_t} \log p_t(\boldsymbol{x}_t|\boldsymbol{y}) = (1 - w)\nabla_{\boldsymbol{x}_t} \log p_t(\boldsymbol{x}_t) + w\nabla_{\boldsymbol{x}_t} \log \hat{p}_t(\boldsymbol{x}_t|\boldsymbol{y}), \tag{10}$$

where $w$ is generally a positive number for smooth control between data consistency and realistic. In the context of the restoration-based likelihood, the data consistency is further exteriorized as restorer intensity to flexibly release the power of the restoration model. Substituting the derived restoration-based likelihood in Eq. 8 enable the posterior sampling from the restorer guidance. The conditional score function in Eq. 10 formally comes to be:

$$\nabla_{\boldsymbol{x}_t} \log p_t(\boldsymbol{x}_t|\boldsymbol{y}) \simeq \eta \boldsymbol{s}_{\theta^*}(\boldsymbol{x}_t, t) - \rho \nabla_{\boldsymbol{x}_t}\|\boldsymbol{x}_t - \sqrt{\bar{\alpha}(t)}\mathcal{R}(\boldsymbol{y})\|_2^2, \tag{11}$$

where we release the strict constrain in Eq. 10 and set the parameters $\eta$ and $\rho$ as harmonic step size for the unconditional prior term and restoration-based likelihood term, considering the complicated balance between restorer intensity and data realistic countered by the diffusion model.

**Related to the classifier-free guidance.** It is worth noting that the prevailing measurement-based likelihood is homologous to the classifier guidance Dhariwal & Nichol (2021), considering the same

| **Algorithm 1** DDPM - Posterior Sampling | **Algorithm 2** DDIM - Posterior Sampling |
|---|---|
| **Require:** $N, \boldsymbol{y}, \eta, \rho, \zeta, \{\tilde{\sigma}_t\}_{t=1}^N, \mathcal{R}(\cdot)$ | **Require:** $N, \boldsymbol{y}, \eta, \rho, \zeta, \{\tilde{\sigma}_t\}_{t=1}^N, \mathcal{R}(\cdot)$ |
| 1: $\boldsymbol{x}_N \sim \mathcal{N}(\sqrt{\bar{\alpha}_N}\boldsymbol{y}, (1-\bar{\alpha}_N)\boldsymbol{I})$ | 1: $\boldsymbol{x}_N \sim \mathcal{N}(\sqrt{\bar{\alpha}_N}\boldsymbol{y}, (1-\bar{\alpha}_N)\boldsymbol{I})$ |
| 2: **for** $t = N-1$ **to** $0$ **do** | 2: **for** $t = N-1$ **to** $0$ **do** |
| 3: $\quad \hat{\boldsymbol{s}} \leftarrow \boldsymbol{s}_\theta(\boldsymbol{x}_t, t)$ | 3: $\quad \hat{\boldsymbol{s}} \leftarrow \boldsymbol{s}_\theta(\boldsymbol{x}_t, t)$ |
| 4: $\quad \hat{\boldsymbol{x}}_{0|t} \leftarrow \frac{1}{\sqrt{\bar{\alpha}_t}}(\boldsymbol{x}_t + (1-\bar{\alpha}_t)\hat{\boldsymbol{s}})$ | 4: $\quad \hat{\boldsymbol{x}}_{0|t} \leftarrow \frac{1}{\sqrt{\bar{\alpha}_t}}(\boldsymbol{x}_t + (1-\bar{\alpha}_t)\hat{\boldsymbol{s}})$ |
| 5: $\quad \boldsymbol{z} \sim \mathcal{N}(\boldsymbol{0}, \boldsymbol{I})$ | 5: $\quad \boldsymbol{z} \sim \mathcal{N}(\boldsymbol{0}, \boldsymbol{I})$ |
| 6: $\quad \boldsymbol{x}'_{t-1} \leftarrow \frac{\sqrt{\alpha_t}(1-\bar{\alpha}_{t-1})}{1-\bar{\alpha}_t}\boldsymbol{x}_t + \frac{\sqrt{\bar{\alpha}_{t-1}}\beta_t}{1-\bar{\alpha}_t}\mathcal{R}(\hat{\boldsymbol{x}}_{0|t})$ $\quad +\tilde{\sigma}_t\boldsymbol{z}$ | 6: $\quad \boldsymbol{x}'_{t-1} \leftarrow -\sqrt{1-\bar{\alpha}_t}\sqrt{1-\bar{\alpha}_{t-1}-\tilde{\sigma}_{t-1}}\hat{\boldsymbol{s}} +$ $\mathcal{R}(\hat{\boldsymbol{x}}_{0|t}) + \tilde{\sigma}_t\boldsymbol{z}$ |
| 7: $\quad \boldsymbol{r}_t \leftarrow \rho\nabla_{\boldsymbol{x}'_{t-1}}\|\boldsymbol{x}'_{t-1} - \sqrt{\bar{\alpha}_t}\mathcal{R}(\boldsymbol{y})\|_2^2$ | 7: $\quad \boldsymbol{r}_t \leftarrow \rho\nabla_{\boldsymbol{x}'_{t-1}}\|\boldsymbol{x}'_{t-1} - \sqrt{\bar{\alpha}_t}\mathcal{R}(\boldsymbol{y})\|_2^2$ |
| 8: $\quad \boldsymbol{m}_t \leftarrow \zeta\nabla_{\boldsymbol{x}'_{t-1}}\|\boldsymbol{x}'_{t-1} - \sqrt{\bar{\alpha}_t}\boldsymbol{y}\|_2^2$ | 8: $\quad \boldsymbol{m}_t \leftarrow \zeta\nabla_{\boldsymbol{x}'_{t-1}}\|\boldsymbol{x}'_{t-1} - \sqrt{\bar{\alpha}_t}\boldsymbol{y}\|_2^2$ |
| 9: $\quad \boldsymbol{x}_{t-1} \leftarrow \eta\boldsymbol{x}'_{t-1} - \boldsymbol{r}_t + \boldsymbol{m}_t$ | 9: $\quad \boldsymbol{x}_{t-1} \leftarrow \eta\boldsymbol{x}'_{t-1} - \boldsymbol{r}_t + \boldsymbol{m}_t$ |
| 10: **end for** | 10: **end for** |
| 11: **return** $\boldsymbol{x}_0$ | 11: **return** $\boldsymbol{x}_0$ |

role of the classifier and the measurement model played in the conditional score function 4. Beyond, we show that the *restorer guidance* is formally analogous to the classifier-free guidance Ho & Salimans (2022) in terms of the likelihood decomposition (Eq. 9). While the difference lies in the conditional prior term $\nabla_{\boldsymbol{x}_t} \log p_t(\boldsymbol{x}_t|\boldsymbol{y})$ assumed in Eq. 4, resulting in the following score:

$$\nabla_{\boldsymbol{x}_t} \log p_t(\boldsymbol{x}_t|\boldsymbol{y}) = (w+1)\nabla_{\boldsymbol{x}_t} \log \hat{p}_t(\boldsymbol{x}_t|\boldsymbol{y}) - w\nabla_{\boldsymbol{x}_t} \log p_t(\boldsymbol{x}_t), \tag{12}$$

which is exactly the classifier-free guidance that sampling from the linear combination of the unconditional score and conditional score estimates. Compared with restorer guidance, the conditional score in Eq. 12 is provided by extra-trained conditional diffusion model, rather than arbitrary off-the-shelf restorers. It also explains why the constrain in Eq. 10 need to be released as the data realistic cannot be guaranteed by the restorer-based likelihood term, compared to the diffusion guidance.

### 3.4 EXTENSION OF THE RESTORER GUIDANCE

The *restorer guidance* of Eq. 11 presents conceptual transition from measurement-based likelihood to restoration-based likelihood ideologically, and we show that it can be further extended to release the great potential of alternative restorers for constructing powerful diffusion solvers. We here provide three major extensions for original *restorer guidance* in the following.

**Step 1: Gradient orientation.** Apart from the measurement-based likelihood that the conditional gradients from $\nabla_{\boldsymbol{x}_t} \log p_t(\boldsymbol{y}|\boldsymbol{x}_t)$ are traced back to the current $\boldsymbol{x}_t$, the likelihood gradients in restorer guidance $\nabla_{\boldsymbol{x}_t} \log p_t(\boldsymbol{x}_t|\boldsymbol{y})$ can be solely dependent on the unconditional diffusion update, in virtue of the opposite probabilistic graphic direction. Therefore, the parallel gradient update in conditional score function can be replaced with serial update for efficient gradient orientation. Let $\boldsymbol{x}'_{t-1}$ denotes the unconditional update of $\boldsymbol{x}_t$, we can rewrite the Eq. 11 as following:

$$\nabla_{\boldsymbol{x}_t} \log p_t(\boldsymbol{x}_t|\boldsymbol{y}) \simeq \eta\boldsymbol{s}_{\theta*}(\boldsymbol{x}_t, t) - \rho\nabla_{\boldsymbol{x}'_{t-1}}\|\boldsymbol{x}'_{t-1} - \sqrt{\bar{\alpha}(t)}\mathcal{R}(\boldsymbol{y})\|_2^2, \tag{13}$$

where we remain the weighting parameter $\sqrt{\bar{\alpha}(t)}$, considering the harmonic step size of the unconditional diffusion model, and Eq. 13 can be approximately regarded as serial update for brevity,

**Step 2: Restorer traveling.** The likelihood in original restorer guidance only involved $\mathcal{R}(\boldsymbol{y})$ for the application of the restoration model, which is insufficient to release the great potential of alternative restorers for powerful solvers. Proceed from this limitation, we show that the restorer can be invoked recursively for optional choice, with the escort of the diffusion model. Besides the guidance provided from restorers, we explicitly apply the restoration model on the one-step denoising result $\hat{\boldsymbol{x}}_{0|t}$ for reliable data consistency, forming the unconditional update of $\boldsymbol{x}'_{t-1}$ in case of DDPM sampling as following:

$$\boldsymbol{x}'_{t-1} \leftarrow \frac{\sqrt{\alpha_t}(1-\bar{\alpha}_{t-1})}{1-\bar{\alpha}_t}\boldsymbol{x}_t + \frac{\sqrt{\bar{\alpha}_{t-1}}\beta_t}{1-\bar{\alpha}_t}\mathcal{R}(\hat{\boldsymbol{x}}_{0|t}) + \tilde{\sigma}_t\boldsymbol{z}, \quad \boldsymbol{z} \sim \mathcal{N}(\boldsymbol{0}, \boldsymbol{I}), \tag{14}$$

where we denote the $\alpha(t)$ as $\alpha_t$ for simplicity, and $\beta_t \triangleq 1 - \alpha_t$, $\tilde{\sigma}_t$ is the reverse diffusion variance. It is worth noting that the explicit restoration of $\hat{\boldsymbol{x}}_{0|t}$ will not hinder the likelihood gradients from the *restorer guidance*, which can be solely dependent on the unconditional update $\boldsymbol{x}'_{t-1}$ (Eq. 13). We

Table 1: Quantitative comparison of solving variational inverse problems with competitive solvers. The baseline results of restorer prototype are in brown. **Bold**: best, underline: second best.

| Method | Image Dehaze | | | | Rain streak removal | | | | Motion Deblur | | | |
|---|---|---|---|---|---|---|---|---|---|---|---|---|
| | PSNR ↑ | SSIM ↑ | FID ↓ | LPIPS ↓ | PSNR ↑ | SSIM ↑ | FID ↓ | LPIPS ↓ | PSNR ↑ | SSIM ↑ | FID ↓ | LPIPS ↓ |
| NAFNet (Chen et al., 2022) | 30.12 | 0.973 | 4.88 | 0.015 | 33.13 | 0.951 | 26.93 | 0.079 | 33.71 | 0.947 | 8.82 | 0.078 |
| MPRNet (Zamir et al., 2021) | 27.33 | 0.962 | 8.46 | 0.023 | **34.95** | 0.959 | 26.86 | 0.073 | 32.66 | 0.936 | 10.98 | 0.089 |
| IR-SDE (Luo et al., 2023) | 24.90 | 0.924 | 9.45 | 0.039 | 34.20 | **0.964** | **10.30** | **0.019** | 30.63 | 0.901 | **6.33** | **0.062** |
| DPS (Chung et al., 2023b) | 17.29 | 0.650 | 58.78 | 0.276 | 23.18 | 0.627 | 142.55 | 0.340 | 24.86 | 0.742 | 83.96 | 0.371 |
| DDNM (Wang et al., 2023) | 12.68 | 0.556 | 31.72 | 0.217 | 12.96 | 0.453 | 178.24 | 0.366 | 25.52 | 0.752 | 60.83 | 0.304 |
| Restorer guidance - *Bayesian* | **30.21** | **0.975** | **4.58** | **0.013** | 33.54 | 0.957 | 25.71 | 0.071 | **34.28** | **0.953** | 7.59 | 0.064 |
| Restorer guidance - *Null-space* | 30.17 | 0.973 | 4.71 | 0.014 | 33.42 | 0.952 | 26.15 | 0.074 | 33.96 | 0.951 | 8.23 | 0.076 |

provide this extension for optional and the lightweight restorers will cause negligible computational burden, compared to the unconditional score model $s_{\theta^*}(x_t, t)$.

**Step 3: Measurement boosting.** The *restorer guidance* presented so far only depend on the information provided from the restoration model, ignoring the original information possessed in the measurement $y$, which prone to lead the suboptimal prototype-biased solving results. To this end, we reformulate the conditional score function in Eq. 11 to incorporate the information across both sides of the restorer. Combining with above two extensions, we have the following complete conditional score function of the *restorer guidance*:

$$\nabla_{x_t} \log p_t(x_t|y) \simeq \eta s_{\theta^*}(x_t, t) - \rho \nabla_{x'_{t-1}} \|x'_{t-1} - \sqrt{\bar{\alpha}_t} \mathcal{R}(y)\|_2^2 + \zeta \nabla_{x'_{t-1}} \|x'_{t-1} - \sqrt{\bar{\alpha}_t} y\|_2^2, \tag{15}$$

where $\zeta$ is a parameter that controls the strength of score derived from the measurement, $\zeta \ll \rho$, and we perform the gradient ascent in this term to boost the performance of the diffusion solver.

We provide the full version of the posterior sampling from the complete conditional score function of the *restorer guidance* with DDPM sampler and DDIM sampler in Algorithm 1 and 2.

## 3.5 APPLICATION OF THE RESTORER GUIDANCE

The *restorer guidance* release the deficiency of the measurement-based likelihood for variational inverse problems, with the acceding of assorted restoration models considering their comprehensive sensitivity to multifarious deterioration process. Aside from this, we show that the *restorer guidance* can further be applied to other cases with promising sample quality and advanced performance.

**Deterioration control.** The step parameter of the restoration-based likelihood provides us the ability to flexibly control the restorer intensity with desired deterioration removal extent; see Fig. 4. Additionally, we show that the deterioration can further be strengthened with simply reversing the gradient directions of the likelihood terms in Eq. 15, resulting in the proficient deterioration control of both sides. The extension of the restorer traveling will be disabled in the case of deterioration control, while the sample realistic in deterioration strengthen can be assured with the diffusion model.

**Out-of-distribution processing.** The *restorer guidance* is capable of handling out-of-distribution deterioration beyond the alternative restorers. Formally, in that case, the conditional gradients provided from the restoration-based likelihood is unreliable, on account of the unstable results of $\mathcal{R}(y)$. We show that through restorer traveling and amplified measurement boosting, the performance of diffusion solvers on out-of-distribution deterioration can be significantly advanced; see Tab. 2 3.

## 4 EXPERIMENTS

We experimentally evaluate our *restorer guidance* on three variational inverse problems, including image dehazing, rain streak removal, and motion deblurring. The evaluated datasets include 500 images in SOTS-Outdoor Li et al. (2018a), 100 images in Rain100L Yang et al. (2017), and 1111 images in GoPro Nah et al. (2017). The unconditional diffusion model is publicly available that pre-trained on ImageNet of size $256 \times 256$ without any finetuning Dhariwal & Nichol (2021). We adopt the DDIM sampler here, and our method can be accomplished within 10 steps for gratified sample quality. The alternative restorers can be selected from various image restoration models that pre-trained on the suggested problem-specific datasets for proficient guidance, including RESIDE-OTS Li et al. (2018a), Rain-combine Zamir et al. (2021), and GoPro Nah et al. (2017) in our exper-

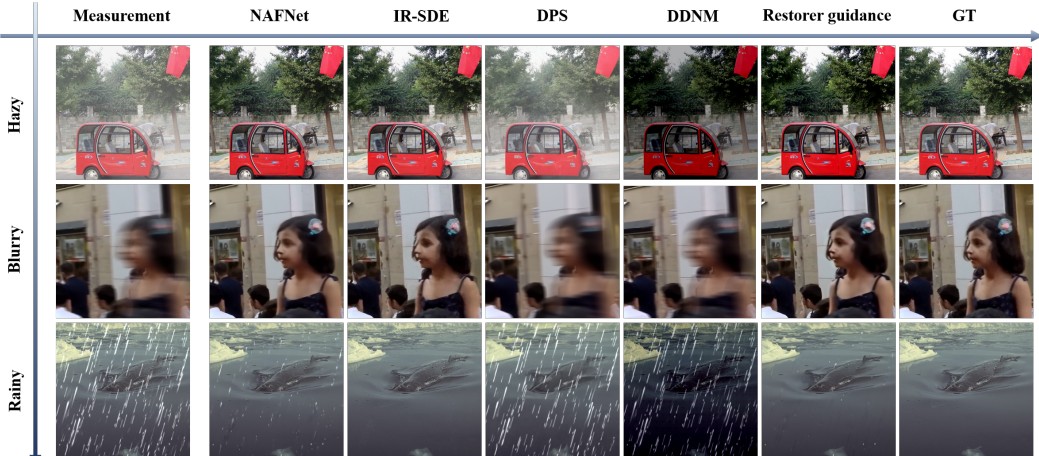

Figure 3: Visual comparison of restorer guidance with other inverse problem solvers on variational deterioration processes, including image dehazing, rain streak removal, and motion deblurring. The restorer prototype is deployed with NAFNet for comparison. Best viewed zoomed in.

iments. We consider the following metrics including the Learned Perceptual Image Patch Similarity (LPIPS) Zhang et al. (2018) and Fréchet Inception Distance (FID) Heusel et al. (2017) for perceptual measurement, and Peak Signal to Noise Ratio (PSNR) and Structural Similarity Index Measure (SSIM) for distortion evaluation. The purpose of experiments is to understand the behavior and potential of the *restorer guidance*, and extend the prevailing diffusion solvers for unprecedented inverse problems beyond the measurement-based likelihood, not necessarily to push the sample quality metrics to state-of-the-art on these benchmarks.

We perform comparison with following methods: Diffusion posterior sampling (DPS) Chung et al. (2023b), denoising diffusion null-space model (DDNM) Wang et al. (2023), Image restoration SDE (IR-SDE) Luo et al. (2023), NAFNet Chen et al. (2022), and MPRNet Zamir et al. (2021). NAFNet and MPRNet are general image restoration backbone, and IR-SDE is a task-specific diffusion solver. DPS and DDNM are measurement-based diffusion solvers for conditional posterior sampling under different frameworks. Considering the inherent deficiency, we parameterize the handcrafted measurement model in DPS and DDNM with network (i.e., NAFNet) for forward variational deterioration process, and the same network architecture is deployed as restorer prototype for comparison.

## 4.1 QUANTITATIVE RESULTS

We show quantitative comparison results in Tab. 1, while the restorer guidance is steadily boosting the performance of the baseline restorer prototype, i.e., NAFNet, on all tasks, regardless of frameworks in Bayesian or range-null space decomposition Wang et al. (2023). This is far beyond exploiting the restoration capability conserved in restorers *losslessly* for visual applications, but rather breaking the upper bound of the restorer for more powerful inverse problem solvers, and also validates the compatibility of the *restorer guidance* with existing unconditional score model. On the other hand, despite the impressive performance the measurement-based methods achieved in solving deterministic inverse problems, the inherent diffciency is manifested when confronted with variational unpredictable deterioration processes. The likelihood derived from the incongruous measurement model and variational contaminated measurements in DPS and DDNM disable the solver behavior completely, compared to the *restorer guidance* which resolved with opposite probabilistic graphic direction of the likelihood. Note we refer the Bayesian version as default in the following.

As presented in Sec. 3.5, the *restorer guidance* is capable of handling out-of-distribution deterioration beyond the incorporated restorers. We present the results of out-of-distribution validation in Tab. 2 and 3 for rain streak removal and motion deblurring, respectively. While the result for image dehazing can be found in Appendix C. In Tab. 2, the comparison methods are trained on Rain100L Yang et al. (2017) while evaluated on Rain100H Yang et al. (2017), differing from the deterioration strength. In Tab. 3, the comparison methods are trained on GoPro Nah et al. (2017) while evaluated on RealBlur-J Rim et al. (2020), differing from the underlying deterioration proto-

type. Observing that the *restorer guidance* is expert at deterioration within the process prototype of the restorer, while releasing the constrain of the deterioration strength (Tab. 2). Moreover, deteriorations beyond the supposed process prototype can also be handled well (Tab. 3), with relatively modest improvement compared to the strength variation. Generally, *restorer guidance* extends the deterministic deterioration process to a cluster of deterioration processes with supposed prototype of the restorer, and enables the sustained release of the restorer capability for augmented cluster space.

Table 2: Out-of-distribution validation of the *restorer guidance*. The comparison methods are trained on Rain100L Yang et al. (2017) while evaluated on Rain100H Yang et al. (2017).

| Methods | PSNR↑ | SSIM↑ | FID↓ | LPIPS↓ |
|---|---|---|---|---|
| NLEDN Li et al. (2018b) | 13.93 | 0.441 | 228.5 | 0.516 |
| *Restorer guidance* | **16.06** | **0.458** | **215.2** | **0.454** |
| PreNet Ren et al. (2019) | 16.48 | 0.565 | 177.8 | 0.401 |
| *Restorer guidance* | **19.00** | **0.587** | **159.9** | **0.352** |

Table 3: Out-of-distribution validation of the *restorer guidance*. The comparison methods are trained on GoPro Nah et al. (2017) while evaluated on RealBlur-J Rim et al. (2020).

| Methods | PSNR↑ | SSIM↑ | FID↓ | LPIPS↓ |
|---|---|---|---|---|
| MPRNet Zamir et al. (2021) | 26.46 | 0.820 | 34.26 | 0.156 |
| *Restorer guidance* | **26.70** | **0.823** | **29.87** | **0.142** |
| Restormer Zamir et al. (2022) | 26.57 | 0.824 | 33.08 | 0.152 |
| *Restorer guidance* | **26.74** | **0.826** | **29.65** | **0.143** |

## 4.2 QUALITATIVE RESULTS AND VISUAL APPLICATIONS

We provide the visual comparison in Fig. 3 to validate the effectiveness and peculiarity of the *restorer guidance* qualitatively. Compared to the baseline restorer, the *restorer guidance* has following merits: (i) Rendering the reconstructed sample with visual pleasing sample quality (e.g., red tricycle), ascribing to the unconditional score model. (ii) Endowing the restoration process with generation capacity that synthesis the nebulous region heuristically (e.g., girl's eye). (iii) Liberating the capability of the restorer continuously for obstinate deterioration (e.g., rain streaks) with ensured data realistic. Compared to measurement-based solvers, the *restorer guidance* is capable to provide more reliable likelihood guidance in variational deterioration process.

In Fig. 4, we provide the visual comparison of *restorer guidance* on out-of-distribution deterioration. The comparison methods are exemplified as PreNet Ren et al. (2019) for rain streak removal and Restormer Zamir et al. (2022) for motion deblurring. The samples drawn from the restorer guidance exhibit the greater robustness to out-of-distribution deterioration, compared to the baseline restorers. The proficient deterioration control achieved by *restorer guidance* is shown in Fig. 5. While one can smoothly controls the restorer intensity via the harmonic step size for desired deterioration extent, and even reverses the restoration process for amplified deterioration. This also provides another perspective for constructing the variational measurement model with reversed restorers rather than handcraft deterministic preferences. Generally, *restorer guidance* provides us a workbench to fabricate the restoration process more flexibly.

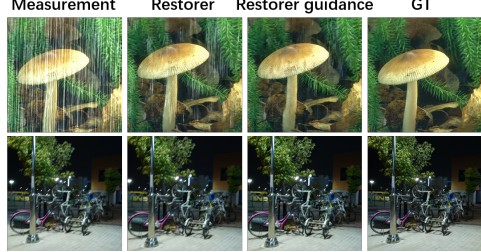

Figure 4: Visual results of out-of-distribution validation of the restorer guidance. First row: Rain100H with PreNet restorer. Second row: RealBlur-J with Restormer restorer.

## 4.3 ABLATION STUDIES

We present the ablation experiments to validate the effectiveness of the suggested extensions attached to the *restorer guidance*. The ablations are performed on problems of rain streak removal and motion deblurring, with reported PSNR and FID metrics. In Tab. 4, we can see that *restorer guidance* attached with extensions further bursts the potential for powerful inverse problem solvers, which is also the key to break the upper bound of the incorporated restorer prototype. Note that the extension of the gradient orientation is adopted as default option to enable the restorer traveling and efficient sampling.

Table 4: Ablation experiments on major extensions attached to the restorer guidance. **RT.**: Restorer traveling. **MB.**: Measurement boosting.

| | | Rain streak removal | | Motion Deblur | |
|---|---|---|---|---|---|
| **RT.** | **MB.** | PSNR ↑ | FID ↓ | PSNR ↑ | FID ↓ |
| ✗ | ✗ | 33.06 | 26.98 | 33.67 | 8.91 |
| ✓ | ✗ | 33.42 | 26.17 | 34.06 | 7.96 |
| ✗ | ✓ | 33.27 | 26.68 | 33.84 | 8.62 |
| ✓ | ✓ | **33.54** | **25.71** | **34.28** | **7.59** |

## 5 RELATED WORK

Image restoration is the classical inverse problem with nondeterministic degradation process imposed on the complete signal, reversing the process with contaminated measurement poses challenges for the solver. Traditional methods incorporated various natural image priors to regularize the underlying solution space, including but not limited to sparse and low-rank prior Lefkimmiatis & Koshelev (2023), dark channel prior He et al. (2010), and deep generative priors Pan et al. (2021); Ulyanov et al. (2018). There methods confined to the deficiency of characterizing the natural image distribution comprehensively, and often resolve the inverse problem with insufficient regularization.

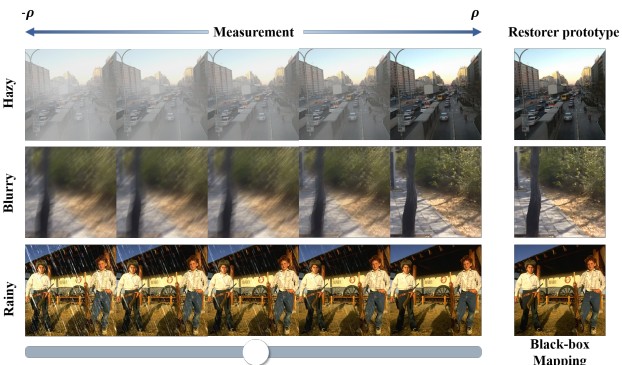

Figure 5: *Restorer guidance* provides us a workbench to fabricate the restoration process more flexible and controllable with proficient deterioration expertise preserved in restorer rather than obstreperous black-box mapping.

Since Sohl-Dickstein et al. (2015) modeling intricate data distribution with inspired non-equilibrium thermodynamics, two successful classes of probabilistic generative models, denoising diffusion probabilistic models (DDPMs) Ho et al. (2020) and score matching with Langevin dynamics (SMLDs) Song & Ermon (2019) have been innovatively developed, which gradually perturb data with noise until tractable distribution and reverse the process with score matching or noise prediction for sampling. Song et al. (2020) amalgamates above two paradigms into a continuous generalized framework with stochastic differential equations. Aside from various generative applications, diffusion models have also been widely appreciated in solving inverse problems. The supervised works typically run the diffusion in the efficient space for deterioration modeling and efficient sampling, including residual space Luo et al. (2023); Yue et al. (2023), frequency space Cao et al. (2022), and latent space Xia et al. (2023). Another line of works adopt diffusion models as regularized priors for zero-shot problem solving, and inject the likelihood for conditional posterior sampling. Pioneer works Chung et al. (2022; 2023b); Song et al. (2023) embrace the Bayes' framework and construct the measurement-based likelihood or generative-based likelihood Chung et al. (2023a); Stevens et al. (2023) for data consistency. Beyond that, Wang et al. (2023) leverage the framework of range-null space decomposition to deliver the balance between realistic and data consistency. However, these methods are confined to the deterministic deterioration process characterized by the measurement model, and impotent to variational unpredictable disturbance in real-world sceneries.

## 6 CONCLUSION AND DISCUSSION

In this work, we proposed the *restorer guidance* for solving variational inverse problems, and shown that the measurement-based likelihood can be replaced with restoration-based likelihood in the opposite probabilistic graphic direction. The *restorer guidance* licencing the patronage of various off-the-shelf restoration models for powerful diffusion solvers, extending the strict deterministic deterioration process to the tolerant cluster process, while attached with extensions further release the great potential of our method. We show that our work is theoretically analogous to the transition from the classifier guidance to classifier-free guidance in the field of inverse problem solver. Extensive experiments illustrate the effectiveness of the *restorer guidance*.

Despite the competitive performance and delightful convenience achieved by *restorer guidance*, it highly depends on the capability of the alternative restorer prototype as baseline performance, which prone to lead the suboptimal prototype-biased solving results. Beyond that, it supposed to incorporate miscellaneous restorer prototypes efficiently with allocated deterioration process to construct the unbiased *restorer guidance* and release the strong dependency from the single prototype in the future. Moreover, *restorer guidance* provides us a workbench to fabricate the restoration process more flexible and controllable with proficient deterioration knowledge, and it supposed to accomplish the interconnected deterioration process with discretionary user inclination in the future.

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

## A  SCORE OF THE RESTORATION-BASED LIKELIHOOD

We here provide another perspective of the score of the restoration-based likelihood, which has been presented in Sec. 3.2. Considering the inevitable deviation between $\boldsymbol{x}_0$ and $\mathcal{R}(y)$, which is so-called restorer bias, the $p(\boldsymbol{x}_0|\boldsymbol{y})$ can be approximated with the following Gaussian:

$$p(\boldsymbol{x}_0|\boldsymbol{y}) \sim \mathcal{N}(\mathcal{R}(\boldsymbol{y}), \boldsymbol{I}), \tag{16}$$

where the mean is obtained from the $\mathcal{R}(\boldsymbol{y})$, with the assumption that the underlying distribution of the mean-reverting error between $\boldsymbol{x}_0$ and $\mathcal{R}(\boldsymbol{y})$ follows the Normal Gaussian, which is exactly the training objective of the restorer $\mathcal{R}$. While the $p(\boldsymbol{x}_t|\boldsymbol{x}_0)$ can be derived from the forward process, which is a linear transform on $\boldsymbol{x}_0$ and adds independent Gaussion noise:

$$\boldsymbol{x}_t = \sqrt{\bar{\alpha}(t)}\boldsymbol{x}_0 + \sqrt{1 - \bar{\alpha}(t)}\epsilon, \qquad \epsilon \sim \mathcal{N}(\mathbf{0}, \boldsymbol{I}). \tag{17}$$

Thus, we have the following approximation to the score, with the consideration that the linear transformation of a Gaussian distribution is still following Gaussian,

$$p(\boldsymbol{x}_t|\boldsymbol{y}) \sim \mathcal{N}(\sqrt{\bar{\alpha}(t)}\mathcal{R}(y), \sqrt{\bar{\alpha}(t)}\sqrt{\bar{\alpha}(t)}\boldsymbol{I} + \sqrt{1 - \bar{\alpha}(t)}\boldsymbol{I}), \tag{18}$$

which can be simplified as:

$$p(\boldsymbol{x}_t|\boldsymbol{y}) \sim \mathcal{N}(\sqrt{\bar{\alpha}(t)}\mathcal{R}(y), \boldsymbol{I}). \tag{19}$$

Note that the above formulation of the $p(\boldsymbol{x}_t|\boldsymbol{y})$ is exactly what we derived in Sec. 3.2, and we here provided another perspective with the broken assumption of the deterministic $p(\boldsymbol{x}_0|\boldsymbol{y})$. The score of the restoration-based likelihood can be written as the following:

$$\nabla_{\boldsymbol{x}_t} \log p(\boldsymbol{x}_t|\boldsymbol{y}) \simeq -\nabla_{\boldsymbol{x}_t} \|\boldsymbol{x}_t - \sqrt{\bar{\alpha}(t)}\mathcal{R}(\boldsymbol{y})\|_2^2, \tag{20}$$

where the same formulation of the likelihood score function is derived, as presented in Sec. 3.2.

## B  RESTORER GUIDANCE IN RANGE-NULL SPACE DECOMPOSITION

Denoising diffusion null-space model (DDNM) Wang et al. (2023) leveraged the framework of range-null space decomposition to delivering the balance between the data consistency and realistic. Considering the noise-free inverse problems first:

$$\boldsymbol{y} = \mathbf{H}\boldsymbol{x}, \tag{21}$$

where, $\boldsymbol{y}$ is the contaminated measurement, $\boldsymbol{x}$ is the original complete signal, $\mathbf{H}$ is the forward measurement model. We adopt the same notations as the what we presented in *restorer guidance* for convenient comparison. Thus, the range-null space decomposition presented that any complete signal $\boldsymbol{x}$ can be decomposed into two parts as following, according to measurement model $\mathbf{H}$:

$$\boldsymbol{x} \equiv \mathbf{H}^\dagger \mathbf{H}\boldsymbol{x} + (\mathbf{I} - \mathbf{H}^\dagger \mathbf{H})\boldsymbol{x}. \tag{22}$$

where $\mathbf{H}^\dagger$ denotes the pseudo-inverse of $\mathbf{H}$, and $\mathbf{I}$ is the identity matrix. Thus, the first part is in the range-space of $\mathbf{H}$ that response for the data consistency, and another part is in the null-space of $\mathbf{H}$ that responsible for data realistic, considering the following formula:

$$\mathbf{H}\boldsymbol{x} \equiv \mathbf{H}\mathbf{H}^\dagger \mathbf{H}\boldsymbol{x} + \mathbf{H}(\mathbf{I} - \mathbf{H}^\dagger \mathbf{H})\boldsymbol{x} \equiv \mathbf{H}\boldsymbol{x} + \mathbf{0} \equiv \boldsymbol{y}. \tag{23}$$

While we can further simplify the formulation in Eq. 24 with Eq. 21 as:

$$\boldsymbol{x} \equiv \mathbf{H}^\dagger \boldsymbol{y} + (\mathbf{I} - \mathbf{H}^\dagger \mathbf{H})\boldsymbol{x}. \tag{24}$$

To solve inverse problems with diffusion models, DDNM performs the decomposition on the one-step denosing result $\boldsymbol{x}_{0|t}$ to enforce the data consistency on range-space and correct the harmonic data realistic on null-space, given by the following formation:

$$\hat{\boldsymbol{x}}_{0|t} = \mathbf{H}^\dagger \boldsymbol{y} + (\mathbf{I} - \mathbf{H}^\dagger \mathbf{H})\boldsymbol{x}_{0|t}, \tag{25}$$

where $\boldsymbol{x}_{0|t}$ is obtained from the Tweedie's formula. Thereby, Eq. 25 delivers the delighted balance between data consistency and data realistic in solving inverse problems.

| **Algorithm 3** DDPM - Null-space sampling | **Algorithm 4** DDIM - Null space Sampling |
|---|---|
| **Require:** $N, \boldsymbol{y}, \{\boldsymbol{\Sigma}_t\}_{t=1}^N, \{\boldsymbol{\Phi}_t\}_{t=1}^N, \mathcal{R}(\cdot)$ | **Require:** $N, \boldsymbol{y}, \{\boldsymbol{\Sigma}_t\}_{t=1}^N, \{\boldsymbol{\Phi}_t\}_{t=1}^N, \mathcal{R}(\cdot)$ |
| 1: $\boldsymbol{x}_N \sim \mathcal{N}(\sqrt{\bar{\alpha}_N}\boldsymbol{y}, (1-\bar{\alpha}_N)\boldsymbol{I})$ | 1: $\boldsymbol{x}_N \sim \mathcal{N}(\sqrt{\bar{\alpha}_N}\boldsymbol{y}, (1-\bar{\alpha}_N)\boldsymbol{I})$ |
| 2: **for** $t = N-1$ **to** 0 **do** | 2: **for** $t = N-1$ **to** 0 **do** |
| 3: $\quad \hat{\boldsymbol{s}} \leftarrow \boldsymbol{s}_\theta(\boldsymbol{x}_t, t)$ | 3: $\quad \hat{\boldsymbol{s}} \leftarrow \boldsymbol{s}_\theta(\boldsymbol{x}_t, t)$ |
| 4: $\quad \boldsymbol{x}_{0\mid t} \leftarrow \frac{1}{\sqrt{\bar{\alpha}_t}}(\boldsymbol{x}_t + (1-\bar{\alpha}_t)\hat{\boldsymbol{s}})$ | 4: $\quad \boldsymbol{x}_{0\mid t} \leftarrow \frac{1}{\sqrt{\bar{\alpha}_t}}(\boldsymbol{x}_t + (1-\bar{\alpha}_t)\hat{\boldsymbol{s}})$ |
| 5: $\quad \hat{\boldsymbol{x}}_{0\mid t} = \boldsymbol{x}_{0\mid t} - \boldsymbol{\Sigma}_t(\boldsymbol{x}_{0\mid t} - \mathbf{R}\boldsymbol{y})$ | 5: $\quad \hat{\boldsymbol{x}}_{0\mid t} = \boldsymbol{x}_{0\mid t} - \boldsymbol{\Sigma}_t(\boldsymbol{x}_{0\mid t} - \mathbf{R}\boldsymbol{y})$ |
| 6: $\quad \boldsymbol{z} \sim \mathcal{N}(\boldsymbol{0}, \boldsymbol{I})$ | 6: $\quad \boldsymbol{z} \sim \mathcal{N}(\boldsymbol{0}, \boldsymbol{I})$ |
| 7: $\quad \boldsymbol{x}_{t-1} \leftarrow \frac{\sqrt{\alpha_t}(1-\bar{\alpha}_{t-1})}{1-\bar{\alpha}_t}\boldsymbol{x}_t + \frac{\sqrt{\bar{\alpha}_{t-1}}\beta_t}{1-\bar{\alpha}_t}\hat{\boldsymbol{x}}_{0\mid t} + \boldsymbol{\Phi}_t \boldsymbol{z}$ | 7: $\quad \boldsymbol{x}_{t-1} \leftarrow -\sqrt{1-\bar{\alpha}_t}\sqrt{1-\bar{\alpha}_{t-1}-\tilde{\sigma}_{t-1}}\hat{\boldsymbol{s}} + \hat{\boldsymbol{x}}_{0\mid t} + \boldsymbol{\Phi}_t \boldsymbol{z}$ |
| 8: **end for** | 8: **end for** |
| 9: **return** $\boldsymbol{x}_0$ | 9: **return** $\boldsymbol{x}_0$ |

Directly applying Eq. 25 to noisy inverse problem leads to the inferior performance, due to the incongruous signal formation, i.e., $\boldsymbol{y} = \mathbf{H}\boldsymbol{x} + \mathbf{n}$, where $\mathbf{n}$ represents the additive Gaussian noise. While the incongruous phenomenon can be formulated as following:

$$\hat{\boldsymbol{x}}_{0\mid t} = \mathbf{H}^\dagger \boldsymbol{y} + (\mathbf{I} - \mathbf{H}^\dagger \mathbf{H})\boldsymbol{x}_{0\mid t} = \boldsymbol{x}_{0\mid t} - \mathbf{H}^\dagger(\mathbf{H}\boldsymbol{x}_{0\mid t} - \mathbf{H}\boldsymbol{x}) + \mathbf{H}^\dagger \mathbf{n}, \tag{26}$$

where $\mathbf{H}^\dagger \mathbf{n}$ is the extra noise introduced in $\hat{\mathbf{x}}_{0\mid t}$, which is undesirable. To this end, DDNM+ modify the Eq. 26 as following to enforce the constrain on the decomposition:

$$\hat{\boldsymbol{x}}_{0\mid t} = \boldsymbol{x}_{0\mid t} - \boldsymbol{\Sigma}_t \mathbf{H}^\dagger(\mathbf{H}\boldsymbol{x}_{0\mid t} - \boldsymbol{y}), \tag{27}$$

where $\boldsymbol{\Sigma}_t$ is set as step size for range-space correction to ensure the data consistency. It is noted that Eq. 29 falls into the the similar formation as measurement-based likelihood in Bayes' framework, which inevitable suffer from the deficiency of the deterministic deterioration process of the measurement model $\mathbf{H}$, and unable to handle variational inverse problems.

Considering the restoration model as $\mathbf{R}$, which supposed to be the exact pseudo-inverse of the underlying variational measurement model $\mathbf{H}$, and vice versa. Therefore, we can rewrite the Eq. 29 as following without any bells and whistles:

$$\hat{\boldsymbol{x}}_{0\mid t} = \boldsymbol{x}_{0\mid t} - \boldsymbol{\Sigma}_t(\mathbf{R}\mathbf{R}^\dagger \boldsymbol{x}_{0\mid t} - \mathbf{R}\boldsymbol{y}). \tag{28}$$

Exploiting the pseudo-inverse trick where $\mathbf{R}\mathbf{R}^\dagger \boldsymbol{x}_{0\mid t} \cong \boldsymbol{x}_{0\mid t}$, we have the following formation:

$$\hat{\boldsymbol{x}}_{0\mid t} = \boldsymbol{x}_{0\mid t} - \boldsymbol{\Sigma}_t(\boldsymbol{x}_{0\mid t} - \mathbf{R}\boldsymbol{y}), \tag{29}$$

which is exactly the principle formation of the *restorer guidance* in range-null space decomposition framework. We provide the complete sampling scheme of *restorer guidance* applied in range-null space decomposition with DDPM sampler and DDIM sampler in Algorithm 3 and 4.

## C   MORE VALIDATION OF OUT-OF-DISTRIBUTION DETERIORATION

We present the results of out-of-distribution validation in Tab. 5 for image dehazing. The comparison methods are trained on RESIDE-OTS Li et al. (2018a) while evaluated on NH-Haze Ancuti et al. (2020), significantly differing from the underlying deterioration prototype. While the same conclusion can be derived from the validation result as presented in Sec. 4.1. The *restorer guidance* handles the deterioration beyond the cluster process of the restorer prototype with relatively modest

Table 5: Out-of-distribution validation of the *restorer guidance*. The comparison methods are trained on RESIDE-OTS Li et al. (2018a) while evaluated on NH-Haze Ancuti et al. (2020).

| Methods | PSNR↑ | SSIM↑ | FID↓ | LPIPS↓ |
|---|---|---|---|---|
| MSBDN Dong et al. (2020) | 12.76 | 0.448 | 299.6 | 0.549 |
| *Restorer guidance* | **12.95** | **0.451** | **291.3** | **0.545** |
| FFANet Qin et al. (2020) | 12.06 | 0.423 | 296.1 | 0.565 |
| *Restorer guidance* | **12.36** | **0.433** | **292.6** | **0.553** |

Figure 6: Qualitative results of out-of-distribution validation on NH-Haze.

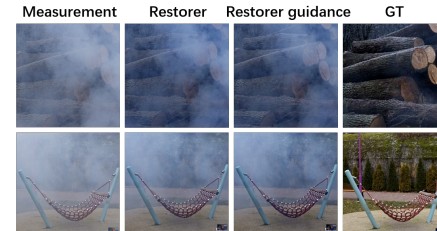

Measurement    Restorer    Restorer guidance    GT

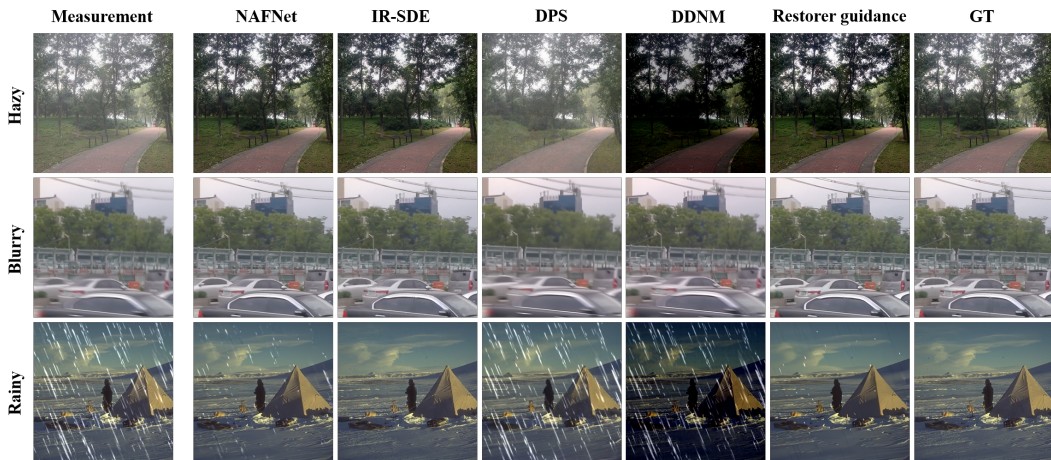

Figure 7: Visual comparison of restorer guidance with other inverse problem solvers on variational deterioration processes, including image dehazing, rain streak removal, and motion deblurring. The restorer prototype is deployed with NAFNet for comparison. Best viewed zoomed in.

improvement, compared to the deterioration strength variation. Fig. 6 presents the qualitative results of the out-of-distribution validation on NH-Haze dataset, where the first row is MSBDN restorer and the second row is FFANet restorer. The *restorer guidance* has slightly improvement compared to the incorporated restorer prototype, albeit the deterioration is far beyond the cluster processes.

## D DETAILED COMPARISON OF VARIOUS MEASUREMENT MODELS

We provide the detailed comparison of various measurement models and the *restorer guidance* in Tab. 6. Considering the measurement model $\mathcal{H}$ with parameters $\phi$, the forward measurement process to the signal $x$ can be formulates as $y = \mathcal{H}(x; \phi)$, where $y$ is the contaminated measurement. Typically, the architecture of the measurement model determines the prototype of the measurement process, and the parameters enable the variability in the surrounding. Handcrafted measurement model Chung et al. (2022; 2023b); Song et al. (2023) restrict to the rigid formulation of $\mathcal{H}$, i.e., the measurement process from $x$ to $y$ is assumed to be the fixed formation such as convolution, addition, and multiplication. While the stationary measurement parameters further restrict the measurement model to the deterministic forward process without any variability. Generated measurement model Chung et al. (2023a); Stevens et al. (2023) remain in the constrain of the rigid formulation of $\mathcal{H}$ with fixed measurement formation, however, the measurement parameters can be jointly estimated with the signal from the generative score model, endowing the variability for handling variational unpredictable measurement process. Parameterized measurement model Fei et al. (2023) extend the handcrafted measurement model with neural networks, breaking the rigid formulation of the measurement process. Albeit the learnable parameters for measurement model, the relaxation for the variability is still restricted, since the non-trivial implementation of the one to many mapping, considering the ill-posed peculiarity of the variational inverse problems. *Restorer guidance* resolves above obstacles with opposite probabilistic graphic direction of the likelihood, compared to the measurement-based methods. The restorer prototype implicitly enable a cluster of measurement processes with desired variability, rather than strict deterministic forward process. Note that except for the handcrafted $\mathcal{H}$, both generated $\mathcal{H}$ and parameterized $\mathcal{H}$ require the extra training of the coupled measurement model, which is time-consuming and inconvenient.

Table 6: Comparison of different measurement models $\mathcal{H}$ and the restorer guidance. The expression involved in the likelihood term are provided. $*_{s_\theta}$ denotes the deterioration parameters are estimated by generative score model. $*_{nn}$ denotes the deterioration process is formulated by network.

| Measurement model | Likelihood term. | Open formula | Variability | Training free |
|---|---|---|---|---|
| Handcrafted $\mathcal{H}$ Chung et al. (2023b) | $\mathcal{H}(x; \varphi) + n$ | ✗ | ✗ | ✓ |
| Generated $\mathcal{H}$ Stevens et al. (2023) | $\mathcal{H}(x; \varphi_{s_\theta}) + n_{s_\theta}$ | ✗ | ✓ | ✗ |
| Parameterized $\mathcal{H}$ Fei et al. (2023) | $\mathcal{H}_{nn}(x; \theta)$ | ✓ | ✗ | ✗ |
| *Restorer guidance* | $\mathcal{R}_{nn}(y; \theta)$ | ✓ | ✓ | ✓ |

# E    ADDITIONAL VISUAL RESULTS

We provide additional visual results in Fig. 7 to further illustrate the effectiveness and the behavior of the *restorer guidance*. More visual results of the proficient deterioration control of the *restorer guidance* are provided in Fig. 8 and 9, where we can fabricate the restoration process more flexible and controllable with proficient deterioration expertise preserved in restorer. It is supposed to accomplish the interconnected deterioration process with discretionary user inclination in the future.

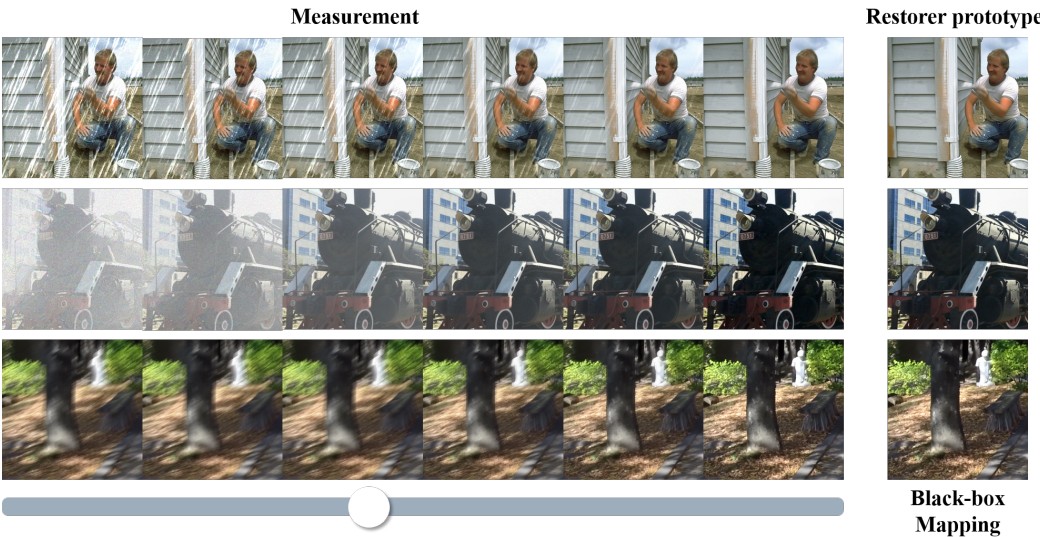

Figure 8: *Restorer guidance* provides us a workbench to fabricate the restoration process more flexible and controllable with proficient deterioration expertise preserved in restorer rather than obstreperous black-box mapping.

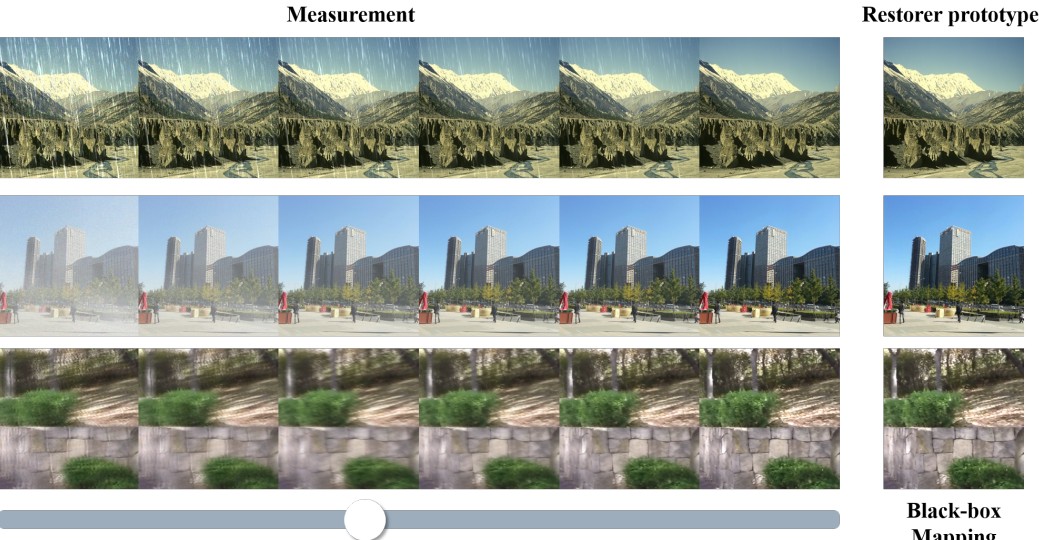

Figure 9: *Restorer guidance* provides us a workbench to fabricate the restoration process more flexible and controllable with proficient deterioration expertise preserved in restorer rather than obstreperous black-box mapping.

