# OpenReview forum: "Restorer Guided Diffusion Models for Variational Inverse Problems"
_ICLR.cc/2024/Conference — ICLR 2024 Conference Withdrawn Submission_

### Official Review · Reviewer_gpTn · 2023-10-13

**Soundness:** 1 poor
**Presentation:** 1 poor
**Contribution:** 1 poor
**Rating:** 1
**Confidence:** 5

**Summary:**

Restorer guided diffusion model, an inverse problem solver which targets blind inverse problems, also covering the case where the parameters of the measurement model is correlated with the underlying signal (e.g. dehazing), is proposed. The method boils down to reverse diffusion constrained to stay close to $\mathcal{R}(\mathbf{y})$, an estimate from a pretrained *restorer*.

**Strengths:**

The paper tackles an important class of inverse problems. Specifically, problems such as dehazing and deraining are inherently different problems from those that are usually dealt with in diffusion model-based inverse problem solvers, e.g. SR, deblurring, inpainting.

**Weaknesses:**

There are several claims in the paper that are misleading, wrong, or not useful.

1-1. Figure 1 does not make a lot of sense to me. The inverse problem is defined to have a measurement likelihood $p(x|y)$, and there cannot be two more classes to these problems. For instance, [1] belongs to the family of blind inverse problems, and you cannot call them *generative*, just because the parameters of the forward model is estimated from a generative model. That has nothing to do with the definition of an inverse problem. It is how you solve it.

1-2. (pg 2, paragraph 1) "However, these methods remain in the paradigm of the measurement-based likelihood": why do you separate (a) and (b) in Fig. 1 then?

1-3. *restoration* is related to $p(x|y)$, while the measurement process is $p(y|x)$. You *cannot* define the measurement process through a restoration model.

2. (pg 2, paragraph 2) "beyond the restriction of deterministic deterioration process without any extra training...": You do need extra training, as you need a *restorer* for guidance. This is not any different from learning different bridges for specific degradation models such as IR-SDE.

3. Eq (7) is not useful. The essence of posterior sampling is to estimate $p(x_0|y)$, which is a component of the integrand. If you are modeling $p(x_0|y)$ with a Gaussian distribution with mean centered on $\mathcal{R}(y)$, then it means that your posterior samples will be the predicted estimates from your restorer plus some AWGN.

4. The method is un-related to classifier-free guidance. In order for CFG to correspond to sharpening the posterior distribution with the parameter $\omega$, the multiplicative constant should remain $1 - \omega$ and $\omega$. Introducing new parameters with $\eta, \rho$ can simply be considered a single parameter by defining $\eta/\rho$. This is no different from how the gradient of the log likelihood is weighted in DPS.

5. Because Eq (7) is not useful, and because of the point made in 3, the method presented in Eq (11) simply boils down to reverse diffusion constrained to be close to $\mathcal{R}(y)$. I can understand that this might boost the performance of the restorer a little bit by leveraging the pre-trained generative prior of the score function, but with extensively more computational burden.

6. Overall, the method is not grounded either theoretically or intuitively.

7. The only really *useful* comparison is against NAFNet and IR-SDE. All other comparisons are not useful. The method should compete against state-of-the-art supervised methods, as it leverages the pretrained restoration model itself.

**References**

[1] Chung, Hyungjin, et al. "Parallel diffusion models of operator and image for blind inverse problems." CVPR 2023.

**Questions:**

1. What is the meaning of *variational* $\mathcal{H}$? This is not a usual term.

2. Section 2.2 last sentence: what does likelihood estimation have to do with the whole paper?

3. Restorer traveling does not make sense to me. If the restoration model is trained such that it can invert the measurement process, what effect would it have when passing in the denoised estimate? What is the reason for calling this restorer *traveling*?

4. Measurement boosting does not make sense to me either. gradient regularization to keep the sample close to $y$ is not useful if your measurment model is not simply Gaussian noise.

5. What do you mean when you state that the measurement model for DPS and DDNM is parametrized with NAFNet? To my understanding this should not be possible. Please provide at least a pseudo-code for this.

---

### Official Review · Reviewer_MTeq · 2023-10-27

**Soundness:** 1 poor
**Presentation:** 1 poor
**Contribution:** 2 fair
**Rating:** 1
**Confidence:** 5

**Summary:**

This work develops a method for using diffusion models for inverse problems beyond deterministic forward deterioration processes. They replace measurement based likelihood with a  restoration-based likelihood  using pretrained restoration models to obtain conditional score function for posterior sampling. This allows diffusion based restoration from non-deterministic degradation processes, such as haze, real world blur and rain.

**Strengths:**

Using a pre-trained diffusion model for non-deterministic degradation processes, such as haze, real world blur and rain has been less explored.

**Weaknesses:**

While the paper proposes to go beyond analytical and deterministic forward operators, it needs pretrained task-specific restoration networks. On the other hand, despite using state of the art supervised task-specific restoration networks, the proposed approach provides only a marginal improvement over this end-to-end trained network.
Even the supposed benefits in terms of out of distribution robustness do not seem to be significant when a strong restoration network such as MPRNet baseline is used.

The intuition behind using \mathcal{R}(xˆ0|t)  in step 6. of algorithm 1 and 2 is not clearly described. It is more intuitive for restoration network to take measurement or an estimate of it as input. Section 3.4 Restorer traveling states this step as optional.

It is not clear why a least squares similarity between a noisy iterate and the degraded measurement (3rd term in equation 15) referred to as measurement boosting should be included.
This implies the measurement is highly similar to the solution as t->1, which is not necessarily true.
The ablation study in section 4.2 shows that including restorer traveling and measurement boosting improves performance. Yet, why these should work is not explained.

Furthermore, there are tasks where x and y have different dimensions, for example  super-resolution, compressed sensing where such terms cannot be used.
_______________________________________________________________________________________

Issues with related work (section 5):
 Ulyanov et al.(2018)  use untrained neural networks as priors, this approach is not same as deep generative priors.

A recent work also uses a pre-trained restoration network in the context of diffusion-based restoration. The authors could discuss this work:
PGDiff: Guiding Diffusion Models for Versatile Face Restoration via Partial Guidance
https://arxiv.org/abs/2309.10810


_______________________________________________________________________________________
Terminology and language used in the paper is strange and not consistent with what is used in the literature.
The language makes it a hard read, requiring several attempts to understand what is meant.
Writing needs a lot of improvement.


The sense in which the term "variational" is used in this paper, is very different from its use in literature. The paper uses this to mean unknown and random measurement process.

"manual destruction" instead of real world blur (last line in page 1)
"realistic" instead of realism (multiple instances throughout the paper)

"which release the deficiency of the deterministic deterioration process with bestowed variational capability" paragraph3 in introduction. It is not clear what this means.

Step 2: Restorer traveling in page 5,  "to release the great potential" --> to realise the great potential

It is not clearly described what is a "cluster process" or "augmented cluster process, such terms are used repeatedly in the paper.

Sample typos/mistakes :
"which similar to the classifier guidance"
"and impotent to variational unpredictable"
", which prone to lead the "
"which prone to lead the suboptimal prototype-biased solving result"
"where we remain the weighting parameter"

**Questions:**

I have some questions regarding the comparisons.

The authors mention that they parameterize the  handcrafted measurement model in DPS and DDNM with a network (i.e., NAFNet). Can the authors clarify if they train a network, (NAFNet) to go from clean samples to corrupted measurements?

As the considered degradations are not deterministic, even if a network is trained to mimic the degradation process, the resulting approximation of the measurement does not necessarily match the original measurement.
How is it handled in the comparisons?

DDNM algorithm also requires a pseudo-inverse in addition to the forward model, how do the authors approximate the
pseudo-inverse of the forward model?

In Fig 3, the output of DPS seems to be still blurry. In the DPS paper, there are results which show that DPS can handle space variant blur. I would suggest the authors to address the tasks already demonstrated in the DPS paper using the proposed method. Similarly it would be desirable to demonstrate the performance of the proposed method on the experiments shown in DDNM paper.

---

### Official Review · Reviewer_kcN8 · 2023-10-30

**Soundness:** 1 poor
**Presentation:** 2 fair
**Contribution:** 2 fair
**Rating:** 3
**Confidence:** 4

**Summary:**

In this work, the authors propose a modified diffusion algorithm for solving inverse imaging problems. Instead of relying solely on the measurement model (implying a "forward" modeling of the inverse problem), the authors rely on a restoration network (implying a "backward" model of the inverse problem). Interestingly, the proposed method allows to weight the importance of the blackbox prior in the reconstruction process, leaving space for finetuning to the user. The authors show the good performance of the proposed method on several difficult inverse problems.

**Strengths:**

- The paper is overall well written.
- The idea of relying on a restoration model instead of a forward measurement model is very original and interesting.

**Weaknesses:**

- Despite overall clarity, the paper contains important shortcuts and tends to strongly dismiss the relevance of concurrent works. These include traditional methods (that are weakly reviewed, e.g. [1,2,3]) or do not mention important recent works as [4, 5].
- Similarly, the paper does not seem to perform fair baselines comparisons.
- While the paper is aimed at general inverse problems, experiments are only performed on very specific inverse problems that are of limited relevance in practice.
- Similarly, the proposed method does only seem to apply when measurements and target images live in the same domain, which is a very limiting assumption in inverse problems and prevents from using this method in most real imaging setups. There is thus a mismatch between the claimed generality of the proposed method and its real applicability setup.


**References**

[1] Bredies, Kristian, Karl Kunisch, and Thomas Pock. "Total generalized variation." SIAM Journal on Imaging Sciences 3.3 (2010): 492-526.

[2] Mallat, Stéphane. A wavelet tour of signal processing. Elsevier, 1999.

[3] Chambolle, Antonin, and Thomas Pock. "A first-order primal-dual algorithm for convex problems with applications to imaging." Journal of mathematical imaging and vision 40 (2011): 120-145.

[4] Romano, Yaniv, Michael Elad, and Peyman Milanfar. "The little engine that could: Regularization by denoising (RED)." SIAM Journal on Imaging Sciences 10.4 (2017): 1804-1844.

[5] Zhu, Yuanzhi, et al. "Denoising Diffusion Models for Plug-and-Play Image Restoration." Proceedings of the IEEE/CVF Conference on Computer Vision and Pattern Recognition. 2023.

**Questions:**

I thank the authors for the interesting paper.

**Major comments**

1. I have troubles understanding where the last term in equation (15) comes from. Could the authors expand on that?
2. Moreover, I find this term problematic as it assumes $\mathbf{y}$ and $\mathbf{x'}_t$ live in the same domain. While this is not a problem for *some* inverse problems, it is an issue for *most* real life inverse problems (phase recovery, MRI, CT). Is the method easily transferable to other inverse problems?
3. I find the overall approach interesting - instead of considering the forward measurement problem, the authors consider trained restoration networks. However, I believe that the way general methods are referred to is too dismisive. For instance, the authors claim: "However, these methods remain in the paradigm of the measurement-based likelihood, and confined to the rigid formulation of the measurement model for signal formation, a.k.a., convolution, addition, and multiplication, with merely estimated deterioration parameters, which inevitably restricts their variational capability for more complicated sceneries." This is a strange sentence: it is very common for the measurement process $\mathcal{H}$ to incorporate random elements that are not expected to be estimated (for instance, in deblurring when $\mathcal{H} = k\*\cdot+n$, the noise $n$ is unknown and this is fine). Furthermore, the phrasing seems to say that "convolution, addition and multiplication" operations that are too simple for real life problems. I believe that the authors try to describe here "linear inverse problems", which is not as general as "nonlinear inverse problems", this is true, but all the problems tackled in this paper are ... linear, and resort simply to convolution, addition, and multiplication! Other complicated problems relying on "addition, multiplication and convolution" include MRI and CT imaging, which cannot be tackled by the proposed method. Did I misunderstand anything, in which case the authors could correct me? The following references are still to be added to the paper [4, 5] (these incorporate the forward measurement equation in the algorithm).
4. The authors claim "In this work, we extend prevailing diffusion solvers for variational inverse problems beyond the restriction
of deterministic deterioration process without any extra training.". However, the authors need to train a model $\mathcal{R}$, am I correct? Furthermore, what do the authors mean by a "variational inverse problem"?
5. After (10), the authors state that "w is generally a positive number for smooth control between data consistency and realistic." Is this equation common in the literature? If yes, could the authors provide some references? (also, the sentence is unfinished)
6. If I understand well, DPS is adapted to the proposed framework. But does this make sense? In particular, while I understand that DPS is not well suited for dehazing as well as deraining, it performs rather well on deblurring - but Figure 7 shows no difference between input images and output images for DPS. Could the authors comment on that?
7. Figure 5 suggests that $\rho$ could be negative, is it the case? I didn't find any statement about that in the paper. Maybe move the arrows around "Measurement" below the figure instead of the slide at the bottom of the image? (these arrow suggest taht we can control the degradation in the measurement)
8. In Table 2 and Table 3, is the restorer used by the authors trained on the same data as the competing methods? This is unclear to me.
9. More generally - although I don't expect the authors to have the time to do this for rebuttal, but maybe for future work - it would be nice to include comparisons with traditional methods on deblurring problems (for instance, using TV priors).


**Minor comments**
1. Why not state at the very beginning of the paper the equation that the authors aim at tackling? Is it $\mathbf{y} = \mathcal{H}(\mathbf{x})+\mathbf{n}$?
2. Figure 1 is misleading. Do generative-based likelihood methods necessarily project a point on a manifold, is this linear? What does it mean to be "invalid" for the "underlying measurement process"?
3. "harmonic step size", is this a common denomination in the literature?
4. Beware, there are several unfinished sentences (see after (10), after (13) for example).
5. "multifarious" --> multivarious
6. "Proceed from this limitation," --> rephrase

**References**

[4] Romano, Yaniv, Michael Elad, and Peyman Milanfar. "The little engine that could: Regularization by denoising (RED)." SIAM Journal on Imaging Sciences 10.4 (2017): 1804-1844.

[5] Zhu, Yuanzhi, et al. "Denoising Diffusion Models for Plug-and-Play Image Restoration." Proceedings of the IEEE/CVF Conference on Computer Vision and Pattern Recognition. 2023.

---

### Official Review · Reviewer_tvMe · 2023-10-30

**Soundness:** 2 fair
**Presentation:** 2 fair
**Contribution:** 2 fair
**Rating:** 3
**Confidence:** 5

**Summary:**

The paper primarily focuses on the application of diffusion models to solve a specific class of  inverse problems, where the degradation model are varying among different images.  It introduces a novel called "restorer guidance,"  which calls some existing pre-trained general restoration model for providing the posterior distribution prior conditioned on the measurement, so as to improve the performance of the application of the diffusion models. Experimental evaluations are carried out on several image restoration tasks to assess the performance of the proposed method.

**Strengths:**

1. The paper addresses an application with practical importance. Any significant progress along this line has its interest from the related community.
2. The paper offers a detailed exposition on diffusion models, facilitating understanding of the basic formulations for those who are not specialists in the area.

**Weaknesses:**

1. There is a disconnect between the paper's stated aim and the approach it employs. While the paper claims to address problems involving varying degradation processes, the methodology it presents essentially relies on invoking a pre-trained model designed for general image recovery to handle  varying degradation. Consequently, the paper lacks a specialized design tailored to address the unique challenges posed by unknown and varying degradation process. In essence, the same protocol  working on  existing image restoration diffusion models by simply running a diffusion-based denoising network to post-process the output from an image restoration method. This calls into question the  novelty of the proposed approach.

2. The paper replies on a pre-trained model for general image recovery, which is already equipped to handle varying degradation. This approach essentially sidesteps the central challenge of providing a conditioned posterior distribution based on measurements, a crucial aspect when extending the problem from image denoising to more complex image recovery tasks that involve a non-trivial null space. Consequently, the paper does not appear to make a substantial contribution to advancing the state of the art in this specific problem domain.

3. The mathematical derivations put forth in the paper primarily address linear inverse problems with known measurement matrices. This focus is incongruent with the complexities of image recovery tasks that involve unknown degradation processes, many of which cannot be formulated as linear inverse problems. Given that these derivations are already well-documented in existing literature, the paper does not  make  novel theoretical contributions.

4. The experiments have many limitations.

4.a. The paper's performance is highly contingent upon  the pre-trained restoration model it employs, referred to as the "restorer." The authors utilize pre-trained models with published demonstration on other tasks such as denoising and super-resolution, it should be noted that while these models claim to be suitable for general image restoration, their performance varies across different tasks. The experiments should be conducted on either the same image tasks showed in the paper they cited. Or adopting  some more universally effective model, such as "Restormer,"

4.b. The paper lacks crucial details, including the datasets used for training both the diffusion and restoration models. The characteristics of these datasets are essential for evaluating the generalization performance of the proposed method, yet this information is  absent.

4.c. This paper is more a applied paper. From this viewpoint, the range of models used for comparison is overly narrow. The experimental setup primarily resembles ablation studies rather than comprehensive evaluations against state-of-the-art methods for each image task. Consequently, the experiments fall short of demonstrating the method's effectiveness in comparison to leading approaches in the field.

**Questions:**

Please see the discussion above.